# The Role of Colistin in the Era of New β-Lactam/β-Lactamase Inhibitor Combinations

**DOI:** 10.3390/antibiotics11020277

**Published:** 2022-02-20

**Authors:** Abdullah Tarık Aslan, Murat Akova

**Affiliations:** 1Department of Internal Medicine, Gölhisar State Hospital, Gölhisar, Burdur 15100, Turkey; aslanabdullahtarik@gmail.com; 2Department of Infectious Diseases and Clinical Microbiology, Faculty of Medicine, Hacettepe University, Ankara 06100, Turkey

**Keywords:** carbapenem resistance, colistin, ceftazidime-avibactam, meropenem-vaborbactam, imipenem-relebactam, ceftolozane-tazobactam, β-lactam/β-lactamase inhibitors

## Abstract

With the current crisis related to the emergence of carbapenem-resistant Gram-negative bacteria (CR-GNB), classical treatment approaches with so-called “old-fashion antibiotics” are generally unsatisfactory. Newly approved β-lactam/β-lactamase inhibitors (BLBLIs) should be considered as the first-line treatment options for carbapenem-resistant *Enterobacterales* (CRE) and carbapenem-resistant *Pseudomonas aeruginosa* (CRPA) infections. However, colistin can be prescribed for uncomplicated lower urinary tract infections caused by CR-GNB by relying on its pharmacokinetic and pharmacodynamic properties. Similarly, colistin can still be regarded as an alternative therapy for infections caused by carbapenem-resistant *Acinetobacter baumannii* (CRAB) until new and effective agents are approved. Using colistin in combination regimens (i.e., including at least two in vitro active agents) can be considered in CRAB infections, and CRE infections with high risk of mortality. In conclusion, new BLBLIs have largely replaced colistin for the treatment of CR-GNB infections. Nevertheless, colistin may be needed for the treatment of CRAB infections and in the setting where the new BLBLIs are currently unavailable. In addition, with the advent of rapid diagnostic methods and novel antimicrobials, the application of personalized medicine has gained significant importance in the treatment of CRE infections.

## 1. Introduction

Antimicrobial resistance (AMR) continues to pose a serious public health threat worldwide, and rates of AMR continue to rise in many parts of the world [1,2]. According to the 2019 Centers for Disease Control and Prevention (CDC) report, >13,000 nosocomial infections and >1000 deaths annually were caused by carbapenem-resistant *Enterobacterales* (CRE) in the United States [2]. Similarly, carbapenem-resistant *Acinetobacter baumannii* (CRAB) and multidrug-resistant (MDR) *Pseudomonas aeruginosa* caused 8500 and 32,000 nosocomial infections and 700 and 2700 deaths, respectively, in the United States, in 2017 [2]. The European Centre of Disease Prevention and Control (ECDC) estimated that approximately 420,000 infections and 18,000 deaths in Europe in 2015 could be attributed to antibiotic-resistant bacteria [3]. Considering the significant burden of disease and limited number of available antimicrobials, the World Health Organization (WHO) listed CRAB, carbapenem-resistant *Pseudomonas aeruginosa* (CRPA), CRE and third generation cephalosporin-resistant *Enterobacterales* as critical priority pathogens for the future research and development of novel antimicrobials [4]. Although there has been an increase in the number of antibiotics that can be used in the treatment of resistant infections in recent years, studies showing the development of resistance to some of these agents are accumulating [5]. Additionally, there is widespread uncertainty about the precise role(s) of new antimicrobials in clinical practice [6,7,8]. Because of the significant differences in the molecular epidemiology of carbapenem-resistant Gram-negative bacteria (CR-GNB) and the lack of new antibiotics in many countries, treatment approaches for infections caused by these pathogens differ significantly worldwide.

The current manuscript reviews the role of colistin and new β-lactam/β-lactamase inhibitors (BLBLIs) for the treatment of CR-GNB infections by addressing features of these molecules, including spectrum of activity, resistance mechanisms, and clinical data on efficacy, safety, and adverse events. In addition, novel BLBLIs which are being currently evaluated in phase 3 randomized controlled trials (RCTs) and personalized treatment approaches for CRE infections were summarized. To achieve the purpose of this review, a through literature search was conducted by using Pubmed/Medline, Web of Science, and Scopus databases without any date restriction. The search was undertaken until December 2021 and only articles published in English were evaluated.

## 2. Colistin

### 2.1. General Features

Colistin has a cationic polypeptide structure and was first discovered as a secondary metabolite of the *Paenibacillus polymyxa* subsp. *colistinus* which naturally lives in the soil [9]. The cationic polypeptide structure of colistin is mainly composed of a cyclic heptapeptide containing a tripeptide side chain acylated at the N terminus by a fatty acid tail [10]. Colistin is classically used as a prodrug, namely colistin methanesulfonate (CMS), which has parenteral and nebulization formulations and is less toxic than colistin sulfate. Although colistin sulfate can be used orally for selective digestive tract decontamination and topically for the treatment of bacterial skin infections, it is not preferred for systemic and aerosolized treatment due to the high risk of nephrotoxicity and bronchoconstriction, respectively [11,12]. CMS can be administered via intravenous, intrathecal, and intraventricular routes. This prodrug is transformed into colistin and several inactive compounds in biological fluids. Colistin is significantly active against most common GNB, including *A. baumannii*, *P. aeruginosa*, *Enterobacterales*, and *Stenotrophomonas maltophilia.* Notably, some Gram-negative species are naturally resistant against colistin, including *Proteus* spp., *Providencia* spp., some *Aeromonas* spp., *Chromobacterium* spp., *Edwardsiella* spp., *Morganella morganii*, *Serratia marcescens*, *Burkholderia mallei*, *Burkholderia cepacia*, *Brucella*, *Legionella*, *Campylobacter*, and *Vibrio cholera*. Additionally, colistin has no activity against anaerobic bacteria [11].

Historically, colistin was widely used as a topical agent for eye and ear infections, and initially employed in the 1950s as an intravenous formulation. In 1959, it was approved by the US Food and Drug Administration (FDA) for the treatment of infectious diarrhea and urinary tract infections caused by GNB. Considering the high risk of nephrotoxicity and neurotoxicity related with colistin usage and the discovery of novel effective and safe antibiotics, clinical use of colistin was largely abandoned in the 1970s. Almost two decades later, in the mid-1990s, colistin re-emerged for the treatment of infections with CR-GNB due to the lack of new antibiotics available to treat these infections [13,14]. Even though the exact antibacterial mechanism(s) of action of colistin is/are still unknown, it is mostly explained by disrupting the integrity of the outer membrane and the resultant leakage of the cytoplasmic content of bacteria due to interaction between the positively charged colistin and the negatively charged phosphate moieties of outer membrane lipids [10]. Other more widely accepted mechanisms of action for colistin are the neutralization of GNB endotoxin, which corresponds to the lipid A portion of lipopolysaccharide (LPS), oxidative damage to bacterial DNA, proteins and lipids through the production of reactive oxygen species, and the inhibition of essential respiratory chain enzymes of GNB (type II NADH-quinone oxidoreductases) [15,16].

### 2.2. Pharmacokinetic/Pharmacodynamic (PK/PD) Properties

The principal PK/PD parameter of colistin is the ratio of the area under the concentration-time curve for free drug from 0 to 24 h to the minimum inhibitory concentration (MIC)(*f*AUC0–24/MIC) [17,18,19]. Only 20–25% of CMS administered is typically converted into active colistin [20]. Therefore, it generally takes >36 h to achieve the target serum concentration, even with a loading dose [20]. Although colistin is efficiently reabsorbed by renal tubules and mostly eliminated in a non-renal way, the urinary concentrations of colistin can reach high levels due to the conversion of CMS (mainly extracted by kidneys) into colistin within the urinary tract [20,21,22]. Furthermore, PK parameters of colistin are subject to substantial interpatient variability, even at a given creatinine clearance [23]. Several PK studies indicated that the parenteral administration of CMS is followed by the slow rise of unbound colistin concentration [24,25]. For this reason, the loading dose of colistin has been considered necessary to avoid therapeutic delays, especially in septic patients [26]. Additionally, the attainment of higher initial serum concentrations has been suggested to reduce the likelihood of exposure to subtherapeutic concentrations of colistin, and thus limit the emergence of resistant or heteroresistant strains [27]. The contemporary guidelines recommend an intravenous loading dose of 300 mg colistin base activity (9 million IU) to reach 2 mg/L steady-state concentration in a patient with an ideal body weight of 75 kg [23]. On the other hand, <40% of patients with normal renal function can achieve >2 mg/L steady-state concentration of colistin, even with a maximally allowed daily dose of 360 mg colistin base activity [23]. Although a 2 mg/L average steady-state concentration of colistin seems to be sufficient for bloodstream and urinary tract infections if MIC value for causative microorganism is <2 mg/L, lower respiratory tract infections are more difficult to treat, and the target serum concentration of colistin (2 mg/L) may be adequate for these infections if colistin MIC is <1 mg/L [28]. A recent systematic review and meta-analysis mostly containing observational studies reported that the administration of a colistin loading dose in patients being treated with high maintenance dosage regimens significantly increased the rate of microbiological eradication, but did not provide any benefit for clinical cure, mortality, or nephrotoxicity risk [29]. The daily dose of colistin should be adjusted according to creatinine clearance and whether the patient receives hemodialysis support [23]. In patients receiving dialysis, an additional dose of colistin corresponding to 10% of the baseline dose is required per hour of dialysis to compensate for loss in dialysis.

### 2.3. Toxicity

Colistin is mainly associated with the increasing risk of neurotoxicity and nephrotoxicity in a dose-dependent manner. Fortunately, both colistin-associated neurotoxicity and nephrotoxicity are generally reversible after cessation of the offending drug [30]. Colistin-associated neurotoxicity may be recognized with paresthesia, weakness, dizziness/vertigo, visual disturbances, confusion, ataxia, neuromuscular blockade, and apnea [31]. The most common neurological side effect is paresthesia being seen in almost one-third of patients [31]. Moreover, neuromuscular blockade or apnea is extremely rare. Colistin-associated nephrotoxicity is significantly augmented when the plasma concentration of colistin exceeds 2.5 mg/L, and is estimated to occur in one-third to one-half of colistin-receiving patients [32,33]. Colistin-associated nephrotoxicity significantly correlates with older age, duration of therapy, and presence of baseline renal dysfunction [34,35]. Conversely, the administration of colistin in combination therapy reduces the risk of colistin-associated acute kidney injury (AKI) [36,37]. Colistin-induced kidney injury can be explained by a multifactorial mechanism in which increased oxidative stress, mitochondrial damage, and impaired tubular epithelial permeability play a critical role [30]. Despite 82% higher incidence of AKI than other antibiotics, the great majority of colistin-associated AKI events are mild and reversible, without a higher rate of mortality or the requirement for renal replacement therapy [37].

### 2.4. Acquired Colistin Resistance Mechanisms in CRE, CRAB, CRPA

#### 2.4.1. CRE

The most common colistin resistance mechanism is the modification of the outer membrane LPS via the addition of cationic molecules, such as L-aminoarabinose and phosphoethanolamine to the LPS. These reactions are mainly conducted by the phosphoethanolamine phosphotransferase enzymes. The *pmrE* gene and *pmrHFIJKLM* operon are responsible for the production of the L-aminoarabinose group and its attachment to the lipid A moiety of the LPS [38,39]. A two-component regulatory system consisting of the enzymes PmrA and PmrB is involved in the addition of phosphoethanolamine and L-aminoarabinose to the LPS [38]. The *pmrA* and *pmrB* gene mutations have been encountered frequently as underlying mechanisms of acquired colistin resistance in *Klebsiella pneumoniae* and *Enterobacter aerogenes* [40,41,42,43]. Similarly, another two-component regulatory system (PhoP and PhoQ) activates the transcription of the *pmrHFIJKLM* operon that is responsible for the addition of L-aminoarabinose to the LPS [44,45]. Several mutations in the *phoP* and *phoQ* genes are culprit mechanisms of acquired resistance to colistin in *K. pneumoniae* [46,47,48]. The *mgrB* gene normally suppresses the expression of the PhoQ-encoding gene, and works as a negative regulator of the PhoPQ two-component system [49]. Therefore, inhibition of the *mgrB* gene results in the increased expression of the *phoPQ* operon, thus leading to the synthesis of L-aminoarabinose responsible for the acquisition of colistin resistance. A wide variety of mutations in the *mgrB* gene leading to colistin resistance have been reported so far, particularly in colistin-resistant *K. pneumoniae* and *Klebsiella oxytoca* strains [50,51,52]. Besides these mechanisms of colistin resistance, the inactivation of the *crrB* (colistin resistance regulation) gene results in the overexpression of the *pmrAB* operon, thus leading to the activation of the *pmrHFIJKLM* operon and of the *pmrC* and *pmrE* genes. As a consequence, phosphoethanolamine and L-aminoarabinose synthesis is activated, and leads to colistin resistance [48]. Lastly, mobile colistin resistance (mcr) genes are carried by plasmids and transferred to various genera of *Enterobacterales*, leading to the horizontal transfer of colistin resistance genes. The *mcr-1* gene was firstly reported from China in 2016, and it was isolated from *Escherichia coli* cultured from a pig [53]. Since then, the *mcr-1* gene has been identified in various bacterial species globally. However, phylogenetic analysis revealed that the *mcr-1* gene likely originated in Chinese livestock in the mid-2000s [54]. To date, 12 different types of *mcr* genes that are carried by different types of plasmids possessing various backbones have been reported. The MCR-1 exerts its effect through the addition of phosphoethanolamine to lipid A, as seen in the previously mentioned chromosomal mutations [53].

#### 2.4.2. CRPA

As with *Enterobacterales*, mutations in the PmrAB and PhoPQ two-component systems have been demonstrated to be associated with acquired colistin resistance in *P. aeruginosa* strains [55,56,57,58,59,60]. Moreover, three other two-component systems have been reported to be responsible for colistin resistance in *P. aeruginosa*, namely, ParRS, ColRS, and CprRS. The ParRS (polymyxin adaptive resistance) two-component system is involved in adaptive resistance to colistin [41,55,61]. The alterations in the ParRS two-component system cause the activation of the *pmrHFIJKLM* operon, and thus leads to the addition of L-aminoarabinose to the LPS. Furthermore, the ColRS and CprRS two-component systems may act through the activation of the *phoQ* gene and/or through other genes that have not yet been identified [61]. Finally, *P. aeruginosa* isolates may acquire resistance to colistin by the overexpression of the outer membrane protein H, which binds to negatively charged phosphate moieties, thus preventing colistin from binding to the LPS, and by trapping colistin in the bacterial capsule [62,63].

#### 2.4.3. CRAB

There are two main mechanisms of colistin resistance in *A. baumannii*. In the first, the cationic groups are added to the LPS by mutations in PmrAB [64,65,66,67,68]. These mutations have been shown to result in the overexpression of the *pmrCAB* operon, leading to phosphoethanolamine synthesis. In the second mechanism, acquired resistance to colistin is the consequence of a complete loss of LPS production through mutations in the lipid A biosynthesis genes, namely, *lpxA*, *lpxC*, and *lpxD* [69].

## 3. Colistin vs. Novel BLBLIs for the Treatment of CR-GNB Infections

### 3.1. Colistin

Comparing the efficacy of colistin with other agents for the treatment of CR-GNB infections is extremely difficult due to the large number of different treatment regimens in the comparator arms, the frequent use of combination regimens in both the colistin arm and the comparator arms, and the suboptimal dosing of colistin in many studies. Numerous studies have revealed that almost half of patients treated with colistin for CR-GNB infections develop AKI, and up to two-thirds of these patients have 30-day or in-hospital mortality [70,71,72,73,74,75,76]. Similarly, poor clinical outcomes (e.g., high clinical failure and prolonged hospital stay) were documented with colistin-based regimens for treating CR-GNB infections [77,78,79,80]. With respect to suboptimal PK/PD indexes, especially in lung, bone and central nervous system, and limited efficacy and increased risk of toxicity (nephrotoxicity and neurotoxicity) pertaining to colistin use, the Clinical and Laboratory Standards Institute (CLSI) has recommended changes to colistin breakpoints. Thus, CLSI removed the susceptibility category of polymyxins and the ‘intermediate’ breakpoint for *Enterobacterales*, *P. aeruginosa*, and *Acinetobacter* spp. was established at ≤2 mg/L, implying unreliable clinical effectiveness, even for isolates with a MIC level of 2 mg/L. This change casts doubt on the use of colistin in the treatment of CR-GNB infections [81].

The daily use of colistin is further complicated by the failure of the routine susceptibility tests to detect colistin susceptibility among GNB. These tests (e.g., disk diffusion test and the automated systems) might identify the significant fraction of isolates as susceptible, when in fact, they are resistant according to the currently recommended broth microdilution method [82]. This has a significant potential in hindering the delivery of appropriate targeted therapy. Some host factors can also limit colistin use in critically ill patients, including obesity, augmented renal clearance, increased volume of distribution, and higher risk of toxicity. As a consequence, the use of colistin in CR-GNB infections should be patient-specific.

#### 3.1.1. Monotherapy vs. Combination Therapy

The role of antibiotic combinations in the treatment of infections caused by CR-GNB is a matter of long-standing debate [83,84]. The potential utility of combination therapy comes from improved effectiveness due to the synergism and prevention of resistance development. The latter is particularly important, because many studies have shown the emergence of resistant or heteroresistant isolates and the regrowth of bacteria after colistin monotherapy [85]. However, combination therapy can increase the likelihood of side effects, costs of antimicrobial treatment and selection pressure, which may facilitate the emergence of antibiotic-resistant organisms. In this context, the outcomes of in vitro assays support the rationale behind antibiotic combinations. These experiments showed that the combination of a carbapenem, fosfomycin, or an aminoglycoside with a polymyxin conferred an additive or synergistic killing effect against *P. aeruginosa* strains. Moreover, among carbapenem-resistant *K. pneumoniae* harboring a wide range of colistin resistance rates, the synergy of colistin with carbapenems, rifampin, and chloramphenicol was demonstrated. Similarly, in vitro studies indicated the synergistic interactions between a polymyxin and a glycopeptide, a carbapenem, tigecycline, or rifampin in CRAB strains [86,87]. However, in real-life conditions, these favorable outcomes cannot be obtained consistently by using combination therapies containing colistin for the treatment of CR-GNB infections [88,89,90].

##### CRAB

Systemic infections caused by CRAB that are major difficult-to-treat resistance phenotypes in most countries lead to disproportionately increased mortality compared to other CR-GNB [91,92]. Despite this fact, the most appropriate antimicrobial therapy for CRAB infections has not yet been defined. In fact, determining the contribution of antimicrobial therapy to final clinical outcomes for CRAB infections is indeed a challenge. This can be explained with several factors. First, the patients with CRAB infections generally have multiple comorbidities and acute medical problems. These factors have a significant effect on clinically relevant outcomes, including all-cause mortality and clinical cure/improvement. Second, particularly for nosocomial pneumonia, it is very difficult to differentiate colonization with CRAB from a real infection. Third, in our daily practice, CRAB infections are often treated with combined antimicrobial regimens, and antimicrobial agents are frequently changed at different stages of treatment. Fourth, since CRAB infections are generally polymicrobial, relative contributions of CRAB versus other bacteria on clinical outcomes are difficult to determine. Furthermore, the efficacy of conventional antimicrobials (e.g., colistin, tigecycline, and aminoglycosides) is limited by unfavorable PK/PD characteristics, increasing resistance rate, and high risk of toxicity. Colistin remains active against CRAB isolates, with an average resistance rate hovering around 20% in the USA [93]. Unfortunately, irresponsible use of colistin, not only in human medicine, but also in veterinary medicine, has led to the emergence of colistin-resistant Gram-negative microorganisms in endemic regions.

The site of infection is crucial in decision-making to use colistin alone or as a part of combination regimens for the treatment of CRAB infections. For lower urinary tract infections (UTIs), because of the aforementioned PK/PD advantageous of colistin, colistin monotherapy can be suggested. In contrast, due to the challenges to achieve effective concentrations in lower respiratory tract after intravenous administration, colistin monotherapy may not be a reliable option for the treatment of pneumonia. To circumvent these concerns, colistin may be preferred in combination treatment, despite the lack of clinical benefit in RCTs [36,94,95,96,97,98]. Moreover, the nebulized form of colistin can be used to reach a higher pulmonary concentration without causing systemic toxicity. Studies with nebulized CMS administration (≥1 MIU) have attained concentrations that exceed the susceptibility breakpoints of CRAB and CRPA [99,100]. However, the vast majority of inhaled colistin (>85%) binds to mucin; this has not been taken into account while determining the free colistin concentrations in any of the previous studies [101]. A meta-analysis comparing a combination of nebulized and intravenous colistin with intravenous colistin alone demonstrated that the combined administration significantly mitigated all-cause mortality (OR 0.69, 95% CI 0.50–0.95) and increased clinical response rates (OR 1.81, 95% CI 1.3–2.53, *p* = 0.0005) [102]. Moreover, there was no increased risk of nephrotoxicity in the combination regimens compared with intravenous administration alone (OR 1.11, 95% CI 0.69–1.80) [102]. However, in a recent retrospective multicenter cohort study, if given with at least one in vitro active intravenous antibiotic, nebulized colistin was found to be associated with lower 14-day clinical failure, but not lower 14-day all-cause mortality in patients with nosocomial pneumonia, due to colistin-susceptible CR-GNB [103]. It should be noted that potential benefits of the combination strategy must be balanced against increased risk of respiratory adverse events from nebulized delivery, especially in hypoxic patients [104]. Overall, given the equivocal intraepithelial penetration of colistin in the lung tissue following intravenous administration and the potential for the emergence of resistance against colistin at concentrations achievable with inhaled colistin (6.73 mg/L, interquartile range 4.8–10.1 mg/L), inhaled therapy in addition to intravenous colistin should be prescribed in combination with another active antibiotic [105].

In a meta-analysis including four RCTs and 14 observational studies, there was no significant difference between colistin monotherapy and combination therapy for *A. baumannii* infections with respect to 28-day mortality and clinical response [106]. However, microbiological eradication was more common in combination therapy arm (OR:0.49; 95% CI, 0.32–0.74; *p*: 0.0009). Consistent with the results of this meta-analysis, the AIDA trial, the largest RCT comparing colistin monotherapy with colistin plus meropenem combination therapy, showed no superiority of combination therapy over monotherapy [36]. As 77% (312/406) of the patients included in this RCT were infected by CRAB, the findings of this trial cannot be applicable for CRE and CRPA infections. In a secondary analysis of the AIDA trial investigating the association between the presence of in vitro synergism and clinical outcomes (i.e., 14-day clinical failure, 14-day and 28-day mortality, and microbiological response), 171 patients with infections caused by CRAB (*n* = 131), CRE (*n* = 37), and CRPA (*n* = 3) were evaluated [107]. In vitro testing (checkerboard assay) found synergism for 73 isolates, antagonism for 20, and additivism/indifference for 78. Consequently, synergism was not protective against 14-day mortality (aOR, 1.09; 95% CI, 0.60–1.96) and 14-day clinical failure (aOR, 0.52; 95% CI, 0.26–1.04). Furthermore, no significant difference was present between the comparison groups for any secondary outcome [107]. This study showed that concentrations of colistin and/or meropenem attained at the site of infection can be lower than those required for in vitro synergism, and the time period of achieving synergistic concentrations at the infection site can be inadequate for effective bacterial killing. Moreover, host–pathogen interactions should be regarded as an important confounder on clinically relevant outcomes. In another secondary analysis of the AIDA trial, the mortality rate was lower among patients infected with colistin-resistant CRAB than in colistin-susceptible strains (42.3% vs. 52.8% at 28 days) [108]. Although this difference did not reach statistical significance, this result may suggest that colistin resistance may lead to significant “fitness-cost” in CRAB strains [109]. In contrast with these observations, infection with colistin-resistant *Klebsiella pneumoniae* carbapenemase (KPC)-producing CRE is significantly associated with higher risk of death [110]. These different findings are most likely derived from biological differences between different bacterial species. Moreover, preliminary findings of the OVERCOME trial (presented in European Congress of Clinical Microbiology and Infectious Diseases) are largely parallel with the results of the AIDA trial [111].

Durante-Mangoni et al. [94] conducted an open-label RCT and found similar mortality rate and length of hospital stay between the colistin-rifampin group and colistin monotherapy group in MDR *A. baumannii* infections. On the other hand, microbiological eradication was higher in the combination treatment arm. Another small-scale study (*n* = 43) supported the results of the previous trial, and showed that both treatment groups had similar clinical efficacy for the treatment of ventilatory associated pneumonia (VAP) [95]. Consistently, Sirijatuphat et al. [96] evaluated colistin monotherapy and colistin plus fosfomycin combination therapy for the treatment of patients with CRAB infections in an open-label RCT. Microbiological response were significantly higher in combination group compared with monotherapy group. However, clinical outcomes (clinical cure and 28-day mortality) did not differ between the two groups. Additionally, the combination therapy consisting of colistin and ampicillin-sulbactam was compared with colistin monotherapy in a small-scale RCT, including 39 patients treated in intensive care unit (ICU) for VAP, caused by CRAB susceptible to both ampicillin-sulbactam and colistin. Although clinical failure was significantly lower in combination therapy, 28-day mortality was similar between the two groups [96]. In a meta-analysis, polymyxin-based therapies had a better clinical response as compared with non-polymyxin-based therapies (OR, 1.99; 95% CI, 1.31 to 3.03), and adverse events were significantly more frequent in polymyxin-based therapies (OR, 4.32; 95% CI, 1.39 to 13.48) [35]. However, since 8 of 11 studies included contain serious risk of bias, the results of this meta-analysis should be evaluated cautiously. In addition, high-dose ampicillin-sulbactam is another alternative as a component of combination therapy containing colistin, and as a monotherapy for moderate to severe and mild CRAB infections, respectively [97,112,113,114,115,116].

##### CRE

Several observational studies investigating bloodstream infections (BSIs) caused by CRE indicated a survival advantage of various combination therapies over monotherapy [117,118,119,120,121]. It is important to note that these studies included highly heterogeneous combination and single-drug regimens that prevent unveiling the clinical efficacy of specific treatment strategies. However, some studies showed higher survival rates if meropenem is included in combination therapies while treating KPC-producing *K. pneumoniae* strains with low MIC against carbapenems [119,120]. Tumbarello et al. conducted a multi-center retrospective cohort study including 661 patients with a wide range of infections (mostly BSIs, *n* = 447) caused by CRE, mostly KPC-producing *K. pneumoniae*; combination therapy harboring at least two in vitro active drugs was associated with significantly lower 14-day mortality. Furthermore, the survival rate was significantly higher when meropenem was given in a combination therapy of infections, due to the isolate with a meropenem MIC ≤8 mg/L [122]. A systematic review and meta-analysis assessing only observational studies indicated an association between the combination of polymyxins with carbapenems and lower mortality and higher survival rate. However, these associations are not strong enough to verify the superiority of the combination therapy over monotherapy because of low quality of evidence [88]. Gutiérrez-Gutiérrez et al. performed a multi-center multinational retrospective cohort study, including patients with clinically significant monobacterial BSIs due to carbapenemase-producing *Enterobacterales* (CPE), recruited from 26 hospitals in ten countries. Overall, 343 (78%) patients were treated with appropriate therapy, which was defined as the administration of at least one in vitro active agent within 5 days of the onset of BSI, and 94 (22%) received inappropriate therapy. Appropriate therapy was associated with lower mortality as compared with inappropriate therapy (38.5% vs. 60.6%; adjusted HR: 0.45; 95% CI, 0.33–0.62). Among those receiving appropriate therapy, the crude mortality rate was similar between those receiving combination therapy and monotherapy (35% vs. 41%; adjusted HR: 1.63; 95% CI, 0.67–3.91). On the other hand, combination therapy was associated with lower mortality than monotherapy only in patients with a high risk of mortality (48% vs. 62%; adjusted HR: 0.56; 95% CI, 0.34–0.91) [123]. In contrast with these data, a large-scale survey being conducted by the participation of physicians from 115 hospitals in 8 countries demonstrated that combination therapy was the preferred treatment approach of BSIs, pneumonia, and central nervous system infections. Monotherapy was more frequently chosen for the treatment of complicated UTIs [124].

##### CRPA

In the current literature, there is a paucity of data comparing monotherapy and combination therapies for CRPA infections. However, both AIDA and OVERCOME trials showed no significant differences between colistin monotherapy and colistin plus meropenem combination regimen in terms of 28-day mortality in the subgroup analysis of patients with CRPA infections [36,111]. Additionally, the number of patients recruited in some retrospective observational studies published so far was very low, and in some of these studies, the results were not adjusted for critical parameters [125,126,127]. As a consequence, there are no convincing data supporting the superiority of colistin combination therapy over monotherapy for the treatment of CRPA infections.

## 4. Novel BLBLIs

Systemic infections with CR-GNB are burdened by high risk of mortality, and represent an urgent threat that needs to be addressed. Due to the unavailability of consolidated first line antimicrobial agents to treat severe infections with CR-GNB, physicians have often employed antibiotics characterized by increased toxicity or suboptimal PK/PD indexes. Despite the increased risk of developing resistance to these antibiotics after exposure, carbapenems have been used frequently in combination regimens for many years. However, in response to these dire circumstances, the antibiotic pipeline against CR-GNB has recently been revived. The in vitro activities of these novel BLBLIs against targeted pathogens are shown in Table 1.

### 4.1. Ceftazidime-Avibactam

Ceftazidime-avibactam (CZA) is the first new-generation BLBLI combination to come to the market and was composed of an old cephalosporin (ceftazidime) and a new generation non-β-lactam β-lactamase inhibitor (avibactam) [128]. CZA can inhibit KPC and OXA-48-like carbapenemases, extended-spectrum beta-lactamases (ESBL) and AmpC beta-lactamases [129]. In addition, its activity against non-carbapenemase-producing CRE strains is excellent, despite the existence of diverse resistance mechanisms [130]. However, the median MICs of KPC-3-producing pathogens are generally higher than those of KPC-2 variants, due to the higher hydrolytic activity of KPC-3 against ceftazidime [131]. CZA also has reliable activity against CRPA strains. In various studies, CZA was active against 67% to 88% of CRPA strains [132,133]. In contrast, the conjunction of ceftazidime with avibactam does not improve its activity against CRAB strains [134]. CZA was approved by the US FDA for complicated urinary tract infections (cUTIs), complicated intrabdominal infections (cIAIs) in 2015, and for hospital-acquired pneumonia (HAP)/VAP in 2018 [135]. It was also licensed by the European Medicines Agency (EMA) for infections due to MDR GNB in adults with limited treatment options. Promising results were reported in studies comparing CZA and other therapies for the treatment of CRE infections. Shields et al. demonstrated more successful clinical outcomes among patients receiving CZA than among those being treated with a variety of combinations, including a carbepenem plus colistin. Furthermore, the risk of nephrotoxicity is lower with CZA compared with other combinations [136]. In a retrospective observational study assessing clinical outcomes of CZA salvage therapy in 138 patients with infections caused by KPC-producing *K. pneumonia*, the administration of CZA (alone or in combination) was the only independent predictor of survival in the multivariate analysis of the cohort, including patients with BSIs (75.4% of all patients). The CZA salvage therapy was also associated with lower 30-day mortality as compared with a matched cohort of patients with BSIs treated with alternative agents (36.5% vs. 55.7%; *p* = 0.005) [137]. The efficacies of CZA and colistin were also compared in a multi-center observational study including 137 patients from the CRACKLE (Consortium on Resistance Against Carbapenems in *Klebsiella* and other *Enterobacteriaceae*) cohort [138]. In this cohort, the CZA arm showed higher probability of better outcomes (64%, 95% CI, 57–71%) and lower 30-day adjusted all-cause hospital mortality (9% vs. 32% respectively, *p* = 0.001) than the colistin arm. Consistently, a meta-analysis assessing three observational cohort studies and one post hoc analysis of an RCT demonstrated significantly higher clinical cure and lower mortality rates with CZA treatment [139]. In parallel with CRE infections, a post hoc analysis of five RCTs and a small number of observational studies supported the effectiveness of CZA in either MDR *P. aeruginosa* or CRPA infections [140,141,142,143,144,145,146]. In a recent Spanish retrospective cohort study, the clinical outcomes of 61 consecutive infection episodes mostly composed of pneumonia and BSIs and caused by MDR *P. aeruginosa* were reported. With CZA treatment (47.5% as a combination therapy), the clinical cure was achieved in 54.1% of the patients by day 14, and the 30-day all-cause mortality rate was 13.1% [147]. To date, no pathogen-directed RCT has been conducted for comparing CZA with the best available therapy (BAT) in CRE and CRPA infections. Furthermore, there is no recorded RCT in ClinicalTrials.gov for CZA. It is also important to highlight that there is no convincing evidence for using CZA in combination therapy in place of monotherapy to achieve better clinical response, higher microbiological eradication, and lower mortality in the treatment of CRE and CRPA infections [137,148,149,150]. Similarly, combination regimens do not confer favorable results over CZA monotherapy in terms of the emergence of resistance against CZA [151]. According to a large-scale pharmacovigilance analysis, CZA appears to be associated with a higher risk of mental status changes and encephalopathy [152]. Additionally, acute pancreatitis was an over-reported unexpected designated medical event with CZA [152].

Regrettably, shortly after introducing CZA into routine use, CZA resistance among three patients infected by ST258 KPC-expressing *K. pneumoniae* strains was observed after 10–19 days of therapy, due primarily to an amino acid alteration (D179Y) within or proximal to the omega loop of the KPC enzyme [153]. Interestingly, the same mutation was able to restore meropenem susceptibility in some strains. However, a potential restoration of meropenem susceptibility with KPC variants is not sustainable, and has uncertain implications in daily practice [154]. To date, numerous mutations in *bla_KPC-3_* and *bla_KPC-2_* genes conferring CZA resistance have been published, and CZA resistance, upon exposure to this antibiotic, may be seen in up to 10% of patients because of these mutations [155,156]. Moreover, an increased copy number of carbapenemase genes impaired outer-membrane permeability, and the presence of a variant penicillin binding protein 3 (PBP3) formed by four amino acid insertion and the acquisition of P162S change in *bla_GES5_* (leading to *bla_GES15_*) may be counted as other relevant resistance mechanisms decreasing CZA susceptibility in CRE and CRPA isolates [157,158,159,160,161,162]. In a recent Greek study, a new plasmid-mediated Vietnamese extended-spectrum β-lactamase (VEB)-25 has been identified as a source of CZA resistance in carbapenem-resistant *K. pneumoniae* strains [163]. Both et al. also showed CTX-M-14-driven CZA resistance among OXA-48-producing *K. pneumoniae* isolates [164]. In another study, the in vitro selection of CZA-resistant OXA-48-producing *K. pneumoniae* mutants was undertaken after a serial transfer approach [165]. The whole genome sequencing analysis of terminal mutants demonstrated changes in efflux pump proteins (e.g., AcrB, AcrD, EmrA, Mdt) and OmpK36 outer membrane protein [160]. Among *P. aeruginosa* isolates, deletions of various sizes in the Ω-loop region of chromosomal AmpC gene can result in CZA resistance by changing the avibactam binding pocket region of AmpC β-lactamases [166]. In addition, the administration of CZA and ceftolozane-tazobactam has a potential to select MDR *P. aeruginosa* strains—producing metallo-beta-lactamases (MBLs) and Pseudomonas-derived cephalosporinase (PDC) variants [167]. Xu et al. also revealed conjugative plasmid-mediated *bla_CMY-172_*-associated CZA resistance in clinical KPC-carrying *K. pneumoniae* strains [168].

CZA has potent in vitro activity against OXA-48-like carbapenemase-producing CRE [169,170]. Consistently, a higher rate of clinical success and a lower rate of mortality in patients treated with CZA (as a monotherapy or combination therapy) compared to other therapies were reported in observational studies, including infections caused by OXA-48-producing *Enterobacterales* [171,172]. Ceftazidime is resistant to the hydrolytic activity of the most common OXA-48 variants. However, some variants vigorously inactivate ceftazidime (e.g., OXA-163, OXA-405) due to their enhanced ESBL activity. Intriguingly, avibactam exhibits less potent inhibitory activity against these OXA-48-like variants [173].

The MBLs or double carbapenamase-producing (i.e., MBLs + serine carbapenemase) CRE have been increasingly encountered worldwide, and the combination of aztreonam with CZA can be employed for the treatment of systemic infections caused by these pathogens. This regimen demonstrates potent in vitro activity against MBL-expressing *Enterobacterales*. In a study, CZA ensures the restoration of aztreonam susceptibility in 86% of MBL-producing *Enterobacterales* [174]. Similarly, in a hollow-fiber infection model of MBL-expressing *Enterobacterales*, the concomitant administration of aztreonam 8 g/day given as 2 h or continuous infusion with CZA provided complete bacterial killing and resistance suppression [175]. Nevertheless, PK studies are required to appreciate drug–drug interactions, leading to PK changes that may have an impact on the efficacy of this combination regimen. Likewise, relevant information is lacking for dose adjustment for specific populations, such as patients with chronic kidney disease and children. Additionally, there are no recommended antimicrobial susceptibility testing methods and clinical susceptibility breakpoints for the CZA–aztreonam combination regimen.

In conclusion, CZA is an excellent choice for treating infections caused by KPC or OXA-48-like carbapenemase-producing CRE. It can also be considered as a second line option after ceftolozane-tazobactam for the treatment of CRPA infections. For the treatment of infections with MBL-expressing CRE, CZA can be combined with aztreonam until the availability of aztreonam-avibactam for daily use. The biggest issue with CZA is the emergence of resistance against this antibiotic, particularly in KPC-producing organisms that are consistently demonstrated in preclinical and post-marketing observational studies. Therefore, these findings raise concerns about whether this drug will continue to be effective in the following years when widely prescribed.

### 4.2. Imipenem-Cilastatin-Relebactam

Relebactam is another BLI with a diazabicyclooctane core which is structurally related to avibactam [176]. It ensures a potent activity against KPC-producing *Enterobacterales* and CRPA, but not against *A. baumannii* [177,178]. In a collection from Europe, imipenem-cilastatin-relebactam susceptibility rate was 98% among KPC-producing *K. pneumoniae* isolates [178]. Likewise, the US collection of KPC-producing strains demonstrated the potent in vitro activity of this antibiotic against KPC producers [179]. Similar to meropenem-vaborbactam, OmpK35 and OmpK36 porin mutations increase the MIC values of imipenem-cilastatin-relebactam among KPC-producing strains. Furthermore, KPC-3 and KPC-2 mutations conferring resistance to CZA do not have any effect on imipenem-cilastatin-relebactam [180,181]. However, some variants of the class A GES-type carbapenemases may confer resistance to this agent [177].

In a small, pathogen-directed, double-blind, phase 3 trial (RESTORE-IMI 1) randomizing patients with VAP, HAP, cIAI, or cUTI due to imipenem-resistant GNB to imipenem-cilastatin-relebactam or imipenem-cilastatin and colistin, 31 met the mMITT criteria [182]. The rate of 28-day clinical response was higher in the imipenem-cilastatin-relebactam (71.4%) group, as compared with imipenem-cilastatin plus colistin (40.0%). Consistently, the 28-day all-cause mortality was lower in patients receiving imipenem-cilastatin-relebactam (9.5%) than those being treated with imipenem-cilastatin plus colistin (30.0%). An antibiotic-associated adverse event is less frequent in patients who received imipenem-cilastatin-relebactam compared with imipenem-cilastatin plus colistin (16.1% vs. 31.3%), including treatment-related nephrotoxicity (10% vs. 56%). A recent case series of 21 patients treated with imipenem-cilastatin-relebactam for mixed types of infections (mostly pneumonia) caused predominantly by MDR *P. aeruginosa* confirmed a high survival rate and a low rate of adverse events with imipenem-cilastatin-relebactam therapy [183]. Imipenem-cilastatin-relebactam is most recently approved BLBLI combination for the treatment of cUTIs, cIAIs, and HAP/VAP [184,185].

Consequently, imipenem-cilastatin-relebactam seems to be an appealing treatment option for KPC-expressing *Enterobacterales* and CRPA infections. However, results from pathogen-directed RCTs are needed to safely prescribe this combination for infections caused by these microorganisms.

### 4.3. Meropenem-Vaborbactam

Meropenem-vaborbactam is composed of an injectable synthetic carbapenem and a boronic acid β-lactamase inhibitor [186]. Meropenem-vaborbactam has an excellent in vitro activity only against class A carbapenemase-producing CRE [187]. Among these strains, MICs were lower for meropenem-vaborbactam than those for CZA [188]. No single KPC mutations have been associated with meropenem-vaborbactam resistance until now [189]. However, the overexpression of AcrAB-TolC efflux pump and/or reduced expression of OmpK37 porin or mutations in OmpK35 and OmpK36 outer membrane porins do elevate meropenem-vaborbactam MIC values [188,189,190,191]. In a phase 3 open-label trial encompassing 72 cases with various CRE infections (e.g., BSIs, cUTIs, HAP or VAP, and cIAIs), the efficacy of meropenem-vaborbactam (2 g/2 g q8h in a 3 h infusion) versus BAT, including CZA monotherapy, was compared. Consequently, meropenem-vaborbactam was found to be associated with significantly higher clinical cure rate and lower 28-day mortality rate, as compared with BAT (66% vs. 33%, *p* = 0.008 and 16% vs. 33%, *p* = 0.03 respectively) [192]. Similarly, a liver transplant patient with bacteremia was successfully treated with meropenem-vaborbactam salvage therapy, despite being infected by a CZA-resistant K. *pneumoniae* with KPC-2 D179Y variant (developed after CZA exposure) [193]. Similarly, in a case report from Italy, a critical patient who received CZA treatment for an UTI a week ago and subsequently developed surgical wound infection and secondary bacteremia was presented. The blood culture and wound swab samples taken from this patient turned out KPC-31-carrying CZA- and cefiderocol-resistant *K. pneumoniae*, and this patient was successfully treated with meropenem-vaborbactam [194]. In a retrospective multi-center cohort study including patients receiving CZA (*n* = 105) and meropenem-vaborbactam (*n* = 26) for the treatment of CRE infections (screened isolates were positive only for *bla_KPC_*), there was no statistically significant difference between the two groups in terms of clinical success (62% vs. 69%; *p* = 0.49) [195]. Additionally, the 30- and 90-day mortality rates were similar between the comparison groups. In this study, combination therapy was more frequently administered in the CZA arm compared to the meropenem-vaborbactam arm (61% vs. 15%; *p* < 0.01). However, a post hoc analysis indicated similar results between CZA monotherapy and meropenem-vaborbactam monotherapy groups. Among patients treated with CZA monotherapy, 20% (3/15) of patients who had a recurrence within 90 days developed resistance against CZA. In contrast, no patients with recurrence in the meropenem-vaborbactam group (*n* = 3) developed resistance against this antibiotic. Furthermore, the three patients with on-therapy CZA resistance received renal replacement therapy and had pneumonia, factors that have previously been reported as risk factors for treatment failure and the development of resistance [149].

In conclusion, meropenem-vaborbactam has reliable activity against KPC-producing *Enterobacterales*, without any activity against other CPE. However, since resistance to CZA has been increasingly observed, meropenem-vaborbactam can be a reasonable treatment alternative for KPC-producing *Enterobacterales*. Nevertheless, more clinical data, particularly pathogen-directed RCT, are needed to appreciate the efficacy of meropenem-vaborbactam in the treatment of KPC-expressing CRE infections. Moreover, active surveillance should be undertaken periodically, since more widespread utilization of meropenem-vaborbactam may lead to the emergence of new resistance mechanisms against this agent.

### 4.4. Ceftolozane-Tazobactam

Ceftolozane is a 3′-aminopyrazolium cephalosporin with potent activity against *P. aeruginosa* strains [196]. Ceftolozane-tazobactam confers better anti-pseudomonal activity than all other commercially available BLBLI combinations, due to its enhanced affinity to the PBPs of *P. aeruginosa* [197]. In large-scale in vitro data (*n* = 1019), ceftolozane-tazobactam has an inhibitory effect against 78% of the CRPA isolates [198]. In another study, 28% of carbapenems-, ceftazidime- and cefepime-resistant isolates were susceptible to CZA, and 53% were susceptible to ceftolozane-tazobactam [199]. In this study, 9% of the ceftolozane-tazobactam-resistant isolates were susceptible to CZA, whereas 36% of the CZA-resistant ones were susceptible to ceftolozane-tazobactam. However, the efficacy of ceftolozane-tazobactam diminishes significantly among isolates collected from European continent, as up to 33% of these isolates typically gain carbapenem resistance phenotype by expressing MBLs or GES-type carbapenemases [200,201,202,203]. Moreover, ceftolozane-tazobactam has less efficacy against *P. aeruginosa* isolates from patients with cystic fibrosis. Among the extensively drug-resistant *P. aeruginosa* strains collected from patients with cystic fibrosis, the in vitro susceptibility rate of ceftolozane-tazobactam ranges from 30% to 54% [204,205]. In addition, ceftolozane-tazobactam was very limited to no activity against ESBL-producing *K. pneumoniae*, CRE, and CRAB strains [206,207,208,209,210]. In response to the results of phase 3 trails demonstrating the safety and efficacy of ceftolozane-tazobactam compared to widely prescribed antibiotics for both cUTIs and cIAIs, the FDA approved ceftolozane-tazobactam for the treatment of these infections in adult patients in December 2014 [211,212,213,214,215]. In addition, ceftolozane-tazobactam was later approved by the FDA for HAP/VAP in 2019. However, there is no pathogen-directed trial comparing ceftolozane-tazobactam and BAT for the treatment of either MDR *P. aeruginosa* or CRPA infections. Ceftolozane displays enhanced activity against constitutively expressed pseudomonal AmpC-, OprD-, and efflux pump-associated resistance mechanisms in *P. aeruginosa* strains [201,216]. Unfortunately, in one study, resistance to ceftolozane-tazobactam has been reported in 14% of MDR *P. aeruginosa* isolates during or after exposure [217]. This is mainly driven by de novo mutations affecting AmpC expression [217]. Consistently, new variants (V213A, E221K, G216R, E221G, and Y223H) of PDC were shown to have an ability to hydrolyze ceftolozane-tazobactam [218]. Additionally, two studies reported overexpression and structural modifications in AmpC variants, resulting in high-level resistance against ceftolozane-tazobactam, specifically in *P. aeruginosa* strains with mutator (PAOMS, Δ*mutS*) backgrounds [219,220]. Since ceftolozane-tazobactam does not have any activity against carbapenemase producers, MBLs-related resistance against ceftolozane-tazobactam can be seen among some CRPA strains [200]. Fraile-Ribot et al. demonstrated that almost 10% of patients developed resistance during the treatment of MDR *P. aeruginosa* infections with ceftolozane-tazobactam [221]. In this study, OXA-14-related (originated from OXA-10 by a single N146S mutation) ceftolozane-tazobactam resistance among MDR *P. aeruginosa* strains was also documented after exposure to ceftolozane-tazobactam [219]. In addition, the same group reported the emergence of resistance against CZA and ceftolozane-tazobactam in MDR *P. aeruginosa* strains expressing OXA-2-derived enzymes designated as OXA-539 and OXA-681 [222,223]. Fournier et al. reported that ceftolozane-tazobactam resistance can be raised from the upregulation of PDC genes due to mutations in the regulator AmpR gene, and changes in the enzymes of the peptidoglycan recycling pathway (AmpD, PBP4 and Mpl). In this study, some previously reported PDC variants with mutations increasing the hydrolytic activity of β-lactamases towards ceftolozane-tazobactam such as F147L, ΔL223-Y226, E247K, N373I were also detected in ceftolozane-tazobactam-resistant *P. aeruginosa* strains [224]. Furthermore, modification in MexCD-OprJ efflux pump and mutations in PBP3 can cause ceftolozane-tazobactam resistance in *P. aeruginosa* strains [225]. Clinically, a lack of adequate source control and failure to take ceftolozane-tazobactam as a prolonged infusion regimen may be associated with the emergence of resistance to this combination therapy, after exposure [226].

In a retrospective multicenter cohort study conducted in the US, 200 patients were allocated in ceftolozane-tazobactam vs. either polymyxins- or aminoglycosides-based regimens for the treatment of drug-resistant *P. aeruginosa* infections [227]. The recruited patients represented severely ill patients with 69% in the ICU and 42% in severe sepsis or septic shock at the onset of infection. VAP constituted 52% of all infections; 7% of patients had bacteremia. In multivariate analysis, treatment with ceftolozane/tazobactam was an independent protective factor against both clinical cures (adjusted odds ratio [aOR], 2.63; 95% confidence interval [CI], 1.31–5.30) and AKI (aOR, 0.08; 95% CI, 0.03–0.22). There was no difference between the groups in terms of in-hospital mortality. In an Italian study with a retrospective multi-center 1:2-matched case-control design, patients with nosocomial pneumonia or BSI due to MDR *P. aeruginosa* were included [228]. Similar to the previous study, patients treated with ceftolozane-tazobactam (*n* = 16) were compared with those receiving polymyxins- or aminoglycosides-based therapies (*n* = 32). There was a trend toward higher 14-day clinical cure rates in ceftolozane-tazobactam arm compared with that of colistin/aminoglycoside arm (81.3% vs. 56.3%; *p* = 0.11). Likewise, a trend favoring ceftolozane-tazobactam was identified for 30-day mortality (18.8% vs. 28.1%; *p* = 0.73). Additionally, an increased risk of AKI (25.0% vs. 0%; *p* = 0.04) was observed in patients treated with colistin/aminoglycoside regimens. In another retrospective study, unadjusted analysis showed that clinical and microbiological cure at day 7 was similar between the patients receiving ceftolozane-tazobactam monotherapy and those treated with ceftolozane-tazobactam plus colistin or an aminoglycoside (66.7% vs. 60%) [229]. Furthermore, no significant difference was present between monotherapy and combination therapy regarding the risk of resistance development against ceftolozane-tazobactam during therapy. A recent multi-center retrospective cohort study assessed the outcomes of ceftolozane-tazobactam therapy for adult immunocompromised patients with MDR *P. aeruginosa* infections (*n* = 69), mainly pneumonia, and followed by wound infections. All-cause 30-day mortality and clinical cure rates were 19% and 68%, respectively [230]. With respect to side effects, clinicians should be prudent for the occurrence of agranulocytosis with ceftolozane-tazobactam, particularly in high-risk patients [152].

As a consequence, ceftolozane-tazobactam is a reasonable option for patients infected by CRPA, with a higher in vitro susceptibility detected for isolates from patients without cystic fibrosis, compared to patients with cystic fibrosis. Nevertheless, the propensity of MDR *P. aeruginosa* isolates to display elevated ceftolozane-tazobactam MIC values is concerning, considering that little progress in the development of new antibiotics covering CRPA has been accomplished.

## 5. Other BLBLIs Currently Evaluated in Phase 3 RCTs

### 5.1. Aztreonam-Avibactam

Aztreonam has the ability to resist hydrolysis via MBLs. Aztreonam, however, is frequently susceptible to hydrolysis by ESBLs, AmpC β-lactamases, and serine carbapenemases (KPCs, and OXA-48-like). As plasmids that contain MBL genes usually also contain genes that express several other β-lactamases, avibactam should be combined with aztreonam to overcome the shortcomings of this antibiotic [231,232,233]. Aztreonam-avibactam provides a broad range of activity against CPE. In line with this fact, Sader et al. showed that the MIC_90_ values for aztreonam-avibactam against KPC producers (*n* = 102), MBL producers (*n* = 59), and OXA-48-like producers (*n* = 57) were ≤0.50 mg/L [233]. Similarly, based on the results of in vitro studies, aztreonam-avibactam is also effective against double carbapenemases (i.e., serine and MBL carbapenemases)-producing *Enterobacterales* [234]. In contrast, aztreonam-avibactam is unlikely to restore the activity of aztreonam against *P. aeruginosa* and *A. baumannii* [235]. Unfortunately, before it is routinely used, a novel resistance mechanism against aztreonam-avibactam via the addition of four amino acids to PBP3 was reported especially in NDM-5-harboring *E. coli* strains [236]. Indeed, the modified PBP3 is not sufficient to cause overt aztreonam-avibactam resistance, however, the co-production of CMY-42 presumably plays a critical role in the attenuation of susceptibility to aztreonam-avibactam [237,238]. Recently, Nordmann et al. demonstrated the same aztreonam-avibactam resistance mechanism, not only in NDM-5-carrying *E. coli* strains, but also in OXA-48 and OXA-181-harboring *E. coli* strains [239]. Additionally, PER-2 and PER-4 cannot be efficiently inhibited by avibactam as compared with other class-A β-lactamases. In line with this fact, CZA and aztreonam-avibactam-resistant PER-2 and PER-4-expressing *Enterobacterales* have been reported in the literature so far [240,241,242,243].

In a recent prospective cohort study including MBLs-expressing *Enterobacterales* BSIs (*n* = 102), aztreonam plus ceftazidime-avibactam was reported to be associated with lower 30-day mortality (HR, 0.37; 95% CI, 0.13–0.74; *p* = 0.01), lower clinical failure at day 14 (HR, 0.30; 95% CI, 0.14–0.65; *p* = 0.002), and shorter length of hospital stay (subdistributional HR, 0.49; 95% CI 0.30–0.82; *p* = 0.007) [244]. However, it should be kept in mind that the presence of a significant inoculum effect among CPE strains may herald the risk of clinical failure with aztreonam-avibactam in systemic infections with high inoculum [245]. A phase III RCT is currently recruiting adult patients with a serious GNB infection, including cIAIs, HAP or VAP; these patients are being randomly allocated to aztreonam-avibactam, with or without metronidazole group, or meropenem, with or without colistin group (ClinicalTrials.gov identifier NCT03329092). Another phase III RCT is undertaken to compare the efficacy of aztreonam-avibactam with BAT on serious infections due to MBL-producing organisms (ClinicalTrials.gov identifier NCT03580044).

As a result, aztreonam-avibactam appears to be an attractive treatment alternative for CRE infections, particularly for patients infected with MBL- or double carbapenemase-expressing pathogens.

### 5.2. Cefepime-Zidebactam

Cefepime was combined with some novel BLBLIs, due to its high potency, its stability against AmpC enzymes, and its chemical structure making it easier to protect from β-lactamases, including some class D carbapenemases (e.g., OXA-48). In addition, cefepime does not have anti-anaerobic activity that may provide an advantage in protection against ‘collateral’ damage [246]. Therefore, several novel cefepime plus BLI combinations were produced, with the aim of targeting a wide range of coverage, including carbapanemeases, ESBLs, and AmpC β-lactamases. For instance, zidebactam is a non-β-lactam bicycloacyl hydrazide BLI with intrinsic β-lactam activity [247]. It can bind to PBP2 and thus demonstrates β-lactam activity against *Enterobacterales*, *P. aeruginosa* and *A. baumannii* [248,249,250,251]. Its spectrum of activity encompasses class A, class C and some class D β-lactamases [247]. However, the inhibition of both PBP2 and PBP3 (primarily by cefepime) ensures the stability of this BLBLI against class A, B, C and (some) D β-lactamases [252,253]. Therefore, the activity of cefepime-zidebactam against MBL-producing pathogens comes from the PBP2 inhibitory effect of zidebactam, rather than its anti-MBL activity [244]. In a recent study from India, four amino acid insertion mutations in PBP3 did not confer resistance against cefepime-zidebactam, even though these mutations (e.g., YRIK, YRIN inserts) significantly reduced the activity of aztreonam-avibactam among MBL-expressing *E. coli* [254].

Two global collections of *Enterobacterales* isolates recovered from clinical samples verified its potent in vitro activity against these isolates, with various resistance mechanisms, including ESBLs, AmpC β-lactamases, and carbapenemases [255,256]. Similarly, Vázquez-Ucha et al. reported the high rate of activity (MIC_50/90_ ≤ 0.5/1 mg/L) of cefepime-zidebactam against CPE isolates (*n* = 400), regardless of carbapenemase type [257]. Among *P. aeruginosa* strains collected in the US (*n* = 19), cefepime-zidebactam MIC_50/90_ was 8/32 mg/L [258]. Based on the results of this study, several resistance mechanisms such as MBLs, efflux pump overexpression, reduced OprD function and AmpC overproduction can be associated with elevated cefepime/zidebactam MIC levels in *P. aeruginosa* strains [258]. In another study conducted in New York City hospitals, overexpressions of AmpC and MexX were reported to be associated with higher MIC levels of cefepime-zidebactam among CRPA clinical isolates [259]. Additionally, the in vitro selection of cefepime-zidebactam-resistant *P. aeruginosa* mutants demonstrated requirements of multiple mutations in genes encoding MexAB-OprM and its regulators, as well as PBP2 and PBP3. These mutations resulted in significant fitness cost among these mutants and the human-simulated regimen of cefepime-zidebactam kept its activity against these mutants in the neutropenic mice lung infection model, despite its high MIC levels (16–64 mg/L) [260]. In parallel with this study, the authors showed that cefepime-zidebactam had good in vivo efficacy against the CRPA murine thigh infection model, despite relatively high MIC levels [261]. In contrast, CRPA isolates with 32 mg/L cefepime-zidebactam MIC value did not meet the in vivo efficacy threshold (1 log_10_ reduction in bacterial burden) in another lung infection model study [262]. Because of these findings, the company producing cefepime-zidebactam offered a clinical breakpoint of ≤16 mg/L or ≤32 mg/L for *P. aeruginosa*. Nevertheless, the clinical efficacy of cefepime-zidebactam is not clear against clinical isolates with MICs that are higher than cefepime susceptibility breakpoint level. For CRAB strains, one study documented the low activity of cefepime-zidebactam that had MIC values lower than the dose-dependent susceptibility breakpoint of cefepime in 34% of the isolates [259]. In line with this study, a recent study confirmed the high rate of resistance of imipenem-non-susceptible *A. baumannii* clinical isolates (*n* = 136) against cefepime-zidebactam (8.1% of susceptibility rate and MIC_50/90_ = 16/32 mg/L) [263]. There is no ongoing or registered phase III RCT for cefepime-zidebactam yet.

### 5.3. Cefepime-Taniborbactam

Taniborbactam is a type of boronic acid BLI, such as vaborbactam. Based on in vitro data, cefepime-taniborbactam has antibacterial activity against Ambler class A, B, C, D enzymes, except IMP. Hamrick et al. reported that taniborbactam restored cefepime activity against all clinical *Enterobacterales* isolates (*n* = 112) and a great majority of *P. aeruginosa* strains (38/41). The MIC_90_ values of these strains were 1 and 4 mg/L, respectively. It corresponds to ≥256- and ≥32-fold increases, respectively, in antibacterial activity, compared to that of cefepime alone [264]. This study showed the potent activity of this combination against *P. aeruginosa* strains, with diverse resistance mechanisms such as PDC variants, OprD mutations, increased MexAB-OprM/MexXY-OprM efflux pump expressions, and KPC, GES, or VIM carbapenemases [264]. In another study, taniborbactam diminished the cefepime MIC ≤ 8/4 mg/L for 93.9% of KPC-producing *Enterobacterales* (62/66) [265]. However, taniborbactam restored the antibacterial activitiy of cefepime among 62.5% (25/40) of NDM-producing *Enterobacterales*, and in none of 13 *blaIMP*-harboring *Enterobacterales* [266]. Similarly, in a recent study including 400 CPE isolates, cefepime-taniborbactam exhibited potent activity against OXA-48- and KPC-producing *Enterobacterales*, and reduced activity against MBL-expressing strains [257]. It should also be noted that cefepime-taniborbactam has reliable activity against strains with high CZA MICs, due to KPC-3 omega-loop variants, including D179Y, V240G, A177E/D179Y, and D179Y/T243M [267]. A global collection of cefepime (*n* = 85) and meropenem non-susceptible (*n* = 143) *P. aeruginosa* isolates indicated that the MIC_50/90_ value of cefepime-taniborbactam against this collection was 8/16 mg/L. Indeed, this combination restored cefepime susceptibility among 71% of cefepime non-susceptible strains and meropenem susceptibility in 85% of meropenem non-susceptible strains at ≤8 mg/L susceptibility breakpoint [268]. In a neutropenic murine thigh infection model study, cefepime-taniborbactam combination (2 g/0.5 g q8h as a 2 h infusion) displayed reliable in vivo efficacy against cefepime-resistant and serine-carbapenemase-producing GNB [269].

Taniborbactam is a reversible inhibitor of serin β-lactamases. In contrast, it acts as a competitive inhibitor against MBLs [264]. Wang et al. demonstrated the emergence of resistance against cefepime-taniborbactam (MIC >8 mg/L) among NDM-5-carrying *E. coli* isolates due to PBP-3 mutations [265]. In an RCT currently underway, cefepime-taniborbactam is being compared with meropenem for the treatment of cUTIs in adults.

### 5.4. Sulbactam-Durlobactam

Sulbactam has intrinsic antimicrobial activity against *A. baumannii* strains through binding to PBP1 and PBP3. Durlobactam is another diazabicyclooctane BLI combined with sulbactam, and has been tested in phase I and phase II trials (ClinicalTrials.gov identifiers NCT03310463, NCT02971423, NCT03303924) [270]. Durlobactam has an enhanced activity against class A, class C, and some class D β-lactamases [271]. In a recent large-scale in vitro susceptibility study, 1722 clinical isolates of *Acinetobacter* spp. were tested, and almost 50% of these strains were resistant to carbapenems. In this study, durlobactam reduced the MIC_90_ values of sulbactam by 32-fold compared to those of sulbactam alone [272]. On the other hand, Seifert et al. reported that 9 out of 246 CRAB strains had sulbactam-durlobactam resistance according to the clinical breakpoint for resistance [273]. Similarly, either the presence of NDM-1 or alterations in PBP3 were demonstrated to result in elevated MIC levels of sulbactam-durlobactam (>4 mg/L) [274]. Zaidan et al. presented a case report depicting a 55-year-old female with septic shock due to nosocomial pneumonia caused by pan-drug resistant *A. baumannii.* In this case, cefiderocol and sulbactam-durlobactam combination provided a sustained clinical response as a salvage therapy [275]. Furthermore, in an ongoing open-label phase 3 RCT (ATTACK trial), the efficacy and safety of sulbactam-durlobactam plus imipenem-cilastatin are being compared with imipenem-cilastatin plus colistin combination therapy for the treatment of HAP/VAP and BSIs caused by *A. baumannii* (ClinicalTrials.gov identifier NCT03894046). The pharmaceutical company that manufactures sulbactam-durlobactam announced on its official website the preliminary results of the ATTACK (Acinetobacter Treatment Trial Against Colistin) trial, which showed positive results with sulbactam-durlobactam treatment compared to colistin plus imipenem-cilastatin [276]. Table 2 shows recommendations for the treatment of CRE, CRPA, and CRAB infections by source of infection.

## 6. Personalized Treatment Approach

Personalized treatment is an innovative multi-step medicinal approach that is used to individualize the management of each patient. It is classically referred to as a method considering patient- and pathogen-related factors that may have an impact on disease outcome and its response to treatment [277]. Although personalized medicine is currently most commonly applied in the field of oncology, it can be relevant for any other discipline. As infections caused by CR-GNB represent a global public health threat worldwide, they should become one of the top priorities for personalized treatment. Furthermore, personalized therapy basically implicates cumbersome procedures that may require a long time to obtain results, and high costs in oncology. However, a personalized approach seems more practical in CRE infections, as laboratory tests are much more affordable, and more rapidly available [278]. Understanding of the carbapenem resistance mechanism(s) has crucial clinical implications, and provides an opportunity to individualized antibiotic therapy. For this purpose, several phenotypic and genotypic commercially available methods can be employed, even though each method has their own intrinsic limitations. In addition, although a robust armamentarium of novel BLBLIs for the treatment of CRE infections has been introduced to the market during the last 5–10 years, there is not yet a ‘perfect’ BLBLI that can kill all types of CRE and fully meet the needs of every patient. In the context of personalized medicine, the clinicians should consider the site of infection, severity and risk factors of infection, the immune status of the patient, local epidemiology, the presence of organ dysfunction, previous infections episodes, and antibiotics used in the treatment of these episodes. The ultimate goal of personalized treatment is the prescription of the most efficient antibiotic regimen, limiting the risk of adverse events and collateral damage. Besides these critical parameters of the personalized treatment approach, the type of carbapenemase enzymes has gained significant importance with the development of new BLBLIs. As each novel BLBLI has a unique spectrum of activity, and the emergence of resistance against some of these molecules has already been demonstrated, antimicrobial regimens should be tailored in each different clinical scenario. Firstly, the type of microorganism and carbapenem resistance mechanism(s) should be identified by rapid diagnostic methods. If the causative microorganism has a carbapenemase activity and carries KPC or OXA-48-like carbapenemase, CZA can be considered in the first-line treatment. However, for MBL-producing pathogens, aztreonam-avibactam seems to be a promising agent. Furthermore, meropenem-vaborbactam and imipenem-cilastatin-relebactam have already been available for KPC-producing *Enterobacterales* infections. As CZA-resistant KPC mutants do not have any impact on these compounds, they can also be offered for CZA-resistant KPC-producing *Enterobacterales* infections. Among novel BLBLIs, cefepime-zidebactam has enhanced in vitro activities against KPC, MBLs, and OXA-48-like carbapenemases. Conversely, cefepime-taniborbactam ensures high in vitro efficacy against KPC and OXA-48-like-harboring CPE. Currently, for *P. aeruginosa* and *A. baumannii*, the variety of molecular resistance mechanisms and the scarcity of effective antibiotic options available significantly limit the feasibility of personalized therapy for infections caused by these species. The progress in the research of new resistance mechanisms and investments for the development of novel antimicrobials will make new avenues for the personalized treatment of CR-GNB infections possible. In addition, it should be illustrated whether the personalized approach improves the safety, quality, and costs of the treatment of CR-GNB infections. In this context, the low number of case reports demonstrated the efficacy of a personalized approach for the treatment of complicated difficult-to-manage infections, and for the prevention of systemic infections in a rectally colonized patient [279,280]. Nonetheless, there is an urgent need to incorporate the personalized medical approach into contemporary RCT designs.

## 7. Conclusions

The infections caused by CR-GNB lead to a dynamic and rapidly evolving crisis, and traditional approaches to optimizing the PK-PD parameters of old antibiotics are frequently insufficient for the effective treatment of these infections. Similarly, old-fashioned last-resort antibiotics confer high toxicity and low efficacy. However, several BLBLIs with activity against CR-GNB have received approval over the past decade, and more are expected in the near future. The administration of these antibiotics as monotherapy versus combination therapy (i.e., combination with aminoglycoside, colistin, etc.) has not been tested rigorously. However, phase III RCTs and some observational studies have consistently reported favorable outcomes when these agents are employed as monotherapy. For this reason, if the causative pathogen is susceptible, these BLBLIs can be used without the routine addition of a second agent, even for systemic infections with high inoculum. Unfortunately, resistance to some of these BLBLIs has already been demonstrated. As new antimicrobials are introduced into routine practice against carbapenem-resistant microorganisms, we will continue to learn more about their efficacy and the tendency of causative microorganisms to develop resistance to these agents.

## Figures and Tables

**Table 1 antibiotics-11-00277-t001:** List of the new β-lactam β-lactamase inhibitors against target carbapenem-resistant Gram-negative bacteria.

New BLBLIs	CPE-KPC	CPE-MBLs	CPE-OXA-48	CRPA (Non-MBL-Producing)	CRAB
Ceftazidime-avibactam	+	−	+	+	−
Imipenem-cilastatin-relebactam	+	−	−	+	−
Meropenem-vaborbactam	+	−	−	−	−
Ceftolozane-tazobactam	−	−	−	+	−
Aztreonam-avibactam	+	+	+	−	−
Cefepime-zidebactam	+	+	+	+	−
Cefepime-taniborbactam	+	+/−	+	+	−
Sulbactam-durlobactam	−	−	−	−	+

+, active; −, not active; Abbreviations: BLBLIs, β-lactam β-lactamase inhibitors; CPE, carbapenemase-producing *Enterobacterales*; KPC, *Klebsiella pneumoniae* carbapenemase; MBLs, metallo- β-lactamases; OXA-48, oxacillinase-48; CRPA, carbapenem-resistant *Pseudomonas aeruginosa*; CRAB, carbapenem-resistant *Acinetobacter baumannii.*

**Table 2 antibiotics-11-00277-t002:** Colistin vs. novel β-lactam β-lactamase inhibitors for the treatment of CR-GNB infections, according to infection site.

Carbapenem-Resistant *Enterobacterales*			
Infection Site	Colistin ^a^	Novel β-Lactam β-Lactamase Inhibitors ^b,c^	References
Bloodstream infection, primary or catheter-related	If novel BLBLIs are unavailable or inactive against causative microorganism, colistin can be preferred in monotherapy or combination therapy, according to the severity of infection	Ceftazidime-avibactam (first line) Meropenem-vaborbactam or imipenem-relebactam (alternative) Ceftazidime-avibactam + Aztreonam (for MBL-producing CRE)	[34,74,77,81,84,103,107,116,117,118,119,123,131,132,133,134,140,143,144,145,146,166,167,177,178,187,188,189,190,239]
Pneumonia	Colistin can be considered only as a combination therapy in case of unavailability of novel BLBLIs or presence of in vitro resistance against these agents Addition of inhaled colistin to existing therapy can be suggested	Ceftazidime-avibactam (first line) Meropenem-vaborbactam or imipenem-relebactam (alternative) Ceftazidime-avibactam + Aztreonam (for MBL-producing CRE)	[34,77,81,84,95,96,98,99,103,107,118,123,132,133,134,140,143,144,145,146,166,167,177,178,187,190]
Intra-abdominal infection	If novel BLBLIs are unavailable or inactive against causative microorganism, colistin can be preferred in monotherapy or combination therapy according to the severity of infection	Ceftazidime-avibactam (first line) Meropenem-vaborbactam or imipenem-relebactam (alternative) Ceftazidime-avibactam + Aztreonam (for MBL-producing CRE)	[77,81,84,118,123,132,134,140,143,144,145,146,166,167,177,178,187,188,190]
Urinary tract infection	Colistin can be considered as a monotherapy in case of unavailability of novel BLBLIs or presence of in vitro resistance against these agents	Ceftazidime-avibactam (first line) Meropenem-vaborbactam or imipenem-relebactam (alternative) Ceftazidime-avibactam + Aztreonam (for MBL-producing CRE)	[34,77,81,84,103,118,123,132,133,134,140,143,144,145,166,167,177,178,187,190]
Central nervous system infection	Colistin can be considered only as a combination therapy in case of unavailability of novel BLBLIs or presence of in vitro resistance against these agents Intrathecal colistin can be added to the combination therapy	Ceftazidime-avibactam (first line) Meropenem-vaborbactam or imipenem-relebactam (alternative) Ceftazidime-avibactam + Aztreonam (for MBL-producing CRE)	[84,118,123,134,146,167,178]
**Carbapenem-Resistant *Pseudomonas aeruginosa* ^d^**			
**Infection Site**	**Colistin**	**Novel β-Lactam β-Lactamase Inhibitors**	
Bloodstream infection, primary or catheter-related	In case of novel BLBLIS are unavailable or inactive against causative microorganism	Ceftolozane-tazobactam (first line) Ceftazidime-avibactam (alternative) Imipenem-relebactam (alternative)	[34,74,81,84,103,107,122,123,140,141,142,146,177,178,222,223,224,225]
Pneumonia	In case of novel BLBLIs are unavailable or inactive against causative microorganism Addition of inhaled colistin to existing therapy can be suggested	Ceftolozane-tazobactam (first line) Ceftazidime-avibactam (alternative) Imipenem-relebactam (alternative)	[34,81,84,95,96,98,99,103,107,122,123,140,141,142,146,177,178,222,223,224,225]
Intra-abdominal infection	In case of novel BLBLIs are unavailable or inactive against causative microorganism	Ceftolozane-tazobactam (first line) Ceftazidime-avibactam (alternative) Imipenem-relebactam (alternative)	[81,84,122,123,140,142,146,177,178,222,224,225]
Urinary tract infection	In case novel BLBLIs are unavailable or inactive against causative microorganism	Ceftolozane-tazobactam (first line) Ceftazidime-avibactam (alternative) Imipenem-relebactam (alternative)	[34,81,84,103,122,123,140,142,146,177,178,222,224,225]
Central nervous system infection	In case novel BLBLIs are unavailable or inactive against causative microorganism Intrathecal colistin can be added to the combination therapy	Ceftolozane-tazobactam (first line) Ceftazidime-avibactam (alternative) Imipenem-relebactam (alternative)	[84,123,178,225]
**Carbapenem-Resistant *Acinetobacter baumannii* ^e^**			
Bloodstream infection, primary or catheter-related	Colistin containing combination regimens (first line) for severe infections Colistin monotherapy (alternative)	No currently available agent Sulbactam-durlobactam is promising	[33,34,74,81,84,86,90,92,102,103,104,107,110,122,123,140,270]
Pneumonia	Colistin containing combination regimens Addition of inhaled colistin to existing therapy can be suggested	No currently available agent Sulbactam-durlobactam is promising	[33,34,81,84,86,90,91,92,93,94,95,96,98,99,102,103,104,107,108,109,110,111,112,121,122,123,140,269,270]
Intra-abdominal infection	Colistin containing combination regimens (first line) for severe infections Colistin monotherapy (alternative)	No currently available agent Sulbactam-durlobactam is promising	[33,81,84,86,90,92,102,104,110,122,123,140]
Urinary tract infection	Colistin monotherapy (first line)	No currently available agent Sulbactam-durlobactam is promising	[33,34,81,84,86,90,92,102,103,110,122,123,140]
Central nervous system infection	Colistin containing combination regimens Intrathecal colistin can be added to the combination therapy	No currently available agent Sulbactam-durlobactam is promising	[33,84,86,92,123]

Abbreviations: BLBLI, β-lactam β-lactamase inhibitors; MBL, metallo- β-lactamases; CRE, carbapenem-resistant *Enterobacterales*; ^a^ No specific combination regimen (i.e., containing at least 2 in vitro active agents) can be recommended; ^b^ No evidence supports combination therapy; ^c^ Aztreonam-avibactam, cefepime-zidebactam, and cefepime-taniborbactam are being assessed in phase III trials; ^d^ There are no compelling data comparing combination therapies with monotherapy; ^e^ There is no specific recommendation for combination regimens. However, colistin-meropenem and colistin-rifampin combinations should be avoided based on available data from randomized, controlled trials.

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
