# Peer review of "The Role of Colistin in the Era of New β-Lactam/β-Lactamase Inhibitor Combinations"

_antibiotics, 2022, doi:10.3390/antibiotics11020277_

Round 1

Reviewer 1 Report

Report

The present manuscript 1600236 title: The Role of Colistin in the Era of New β-Lactam/β-Lactamase 2 Inhibitor Combinations highlighted the clinical importance of Newly approved β-lactam/β-lactamase inhibitors (BLBLIs) versus colistin against carbapenem-resistant 10 Gram-negative bacteria (CR-GNB). The topic of this research is very interesting and important from the medical point of view particularly in the era of searching new approaches for combating multidrug resistant (MDR)-pathogens. The review is well written and properly organized and contained very important medical information with help in guidance the physician for the roper choice of the antimicrobial against the WHO priority pathogens. However, I have certain minor comments that should be considered before accepting this work for publications, these include:

  1. Abbreviations should be written in full sentence in the first mention and the abbreviation should be used in the whole manuscript thereafter (Example: line 330 KPC (Klebsiella pneumoniae carbapenemase) and L 198, Acinetobacter baumannii was written as abbreviated Acinetobacter baumannii however, it was rewritten in L 305 as full. All abbreviations should be sued consistently in the whole manuscript)
  2. L135, “The most common neurological side effect is paresthesia being seen in almost 135 one-third of patients” needs citation of the proper reference(s).
  3. The author should discuss the List of the novel new β-lactam/β-lactamase inhibitors against target carbapenem-resistant Gram-negative bacteria in the text with the same order they mentioned in table 1. This is important to help the readers to correlate the antimicrobial activities (delineated in Table 1) with those mentioned in the text.
  4. In the footnote of table1, the authors should define the meaning of (+) and (-) signs.
  5. A list of abbreviations should be included before reference section.
  6. Some important and relevant recent literature that should be included in the manuscript in their relevant position:

  1. Doremus C, Marcella SW, Cai B, Echols RM. Utilization of Colistin Versus β-Lactam and β-Lactamase Inhibitor Agents in Relation to Acute Kidney Injury in Patients with Severe Gram-Negative Infections. Infect Dis Ther. 2021 Nov 3:1–13. doi: 10.1007/s40121-021-00556-x. Epub ahead of print. PMID: 34731456; PMCID: PMC8564277.
  2. Kamel NA, Elsayed KM, Awad MF, Aboshanab KM, El Borhamy MI. Multimodal Interventions to Prevent and Control Carbapenem-Resistant Enterobacteriaceae and Extended-Spectrum β-Lactamase Producer-Associated Infections at a Tertiary Care Hospital in Egypt. Antibiotics (Basel). 2021 Apr 30;10(5):509. doi: 10.3390/antibiotics10050509. PMID: 33946253; PMCID: PMC8146387.
  3. Monogue ML, Sakoulas G, Nizet V, Nicolau DP. Humanized Exposures of a β-Lactam-β-Lactamase Inhibitor, Tazobactam, versus Non-β-Lactam-β-Lactamase Inhibitor, Avibactam, with or without Colistin, against Acinetobacter baumannii in Murine Thigh and Lung Infection Models. Pharmacology. 2018;101(5-6):255-261. doi: 10.1159/000486445. Epub 2018 Mar 13. PMID: 29533955.
  4. Elshamy AA, Saleh SE, Alshahrani MY, Aboshanab KM, Aboulwafa MM, Hassouna NA. OXA-48 Carbapenemase-Encoding Transferable Plasmids of Klebsiella pneumoniaeRecovered from Egyptian Patients Suffering from Complicated Urinary Tract Infections. Biology (Basel). 2021 Sep 9;10(9):889. doi: 10.3390/biology10090889. PMID: 34571766; PMCID: PMC8469419.
  5. Karaoglan I, Zer Y, Bosnak VK, Mete AO, Namiduru M. In vitro synergistic activity of colistin with tigecycline or β-lactam antibiotic/β-lactamase inhibitor combinations against carbapenem-resistant Acinetobacter baumannii. J Int Med Res. 2013 Dec;41(6):1830-7. doi: 10.1177/0300060513496172. PMID: 24265334.

.

Reviewer 2 Report

The paper by Aslan and Akova reviews the role of colistin alone and in combination with new beta-lactam/beta-lactamase inhibitor combinations.  The review is extensive in its overview of the field.  Some minor points need to be addressed.

  1. You have some good statistics for the United States to start your paper, but you should include some statistics and referencing for elsewhere in the world as well.
  2. Line 29 add an a before serious.
  3. Line 49 change to "current manuscript reviews".
  4. Line 61 change to first.
  5. Lines 65 -66 change to "prodrug, namely colistin methanesulfonate (CMS), that".
  6. Line 84 change from 2 to two.
  7. Line 90 change most to more.
  8. Line 295 change underlined to noted.
  9. Line 303 change 4 to four.
  10.  Rewrite line 308.
  11. Italicize gene names in lines 477, 478, and 482.
  12. Line 481 add and before presence.
  13. In vitro is italicized whenever it is used.
